# Association of miR-146a-5p and miR-21-5p with Prognostic Features in Melanomas

**DOI:** 10.3390/cancers16091688

**Published:** 2024-04-26

**Authors:** Maria Naddeo, Elisabetta Broseghini, Federico Venturi, Sabina Vaccari, Barbara Corti, Martina Lambertini, Costantino Ricci, Beatrice Fontana, Giorgio Durante, Milena Pariali, Biagio Scotti, Giulia Milani, Elena Campione, Manuela Ferracin, Emi Dika

**Affiliations:** 1IRCCS Azienda Ospedaliero-Universitaria di Bologna, 40126 Bologna, Italy; maria.naddeo2@unibo.it (M.N.); elisabett.broseghin2@unibo.it (E.B.); 2Department of Medical and Surgical Sciences (DIMEC), University of Bologna, 40126 Bologna, Italy; federico.venturi12@unibo.it (F.V.); mlambertini@hotmail.it (M.L.); costantino.ricci4@unibo.it (C.R.); beatrice.fontana10@unibo.it (B.F.); giorgio.durante3@unibo.it (G.D.); 3Oncologic Dermatology Unit, IRCCS Azienda Ospedaliero-Universitaria di Bologna, 40126 Bologna, Italy; sabina.vaccari@aosp.bo.it (S.V.); biagio.scotti@studio.unibo.it (B.S.); giuliamilani1998@gmail.com (G.M.); 4Division of Pathology, IRCCS Azienda Ospedaliero-Universitaria di Bologna, 40126 Bologna, Italy; barbara.corti@aosp.bo.it; 5Pathology Unit, Ospedale Maggiore, 40133 Bologna, Italy; 6Center for Applied Biomedical Research, S. Orsola-Malpighi University Hospital, 40126 Bologna, Italy; milena.pariali@aosp.bo.it; 7Dermatologic Unit, Department of Systems Medicine, University of Rome Tor Vergata, 00133 Rome, Italy; elena.campione@uniroma2.it

**Keywords:** melanoma, histopathology, miRNA, Breslow thickness, ulceration, prognosis

## Abstract

**Simple Summary:**

Cutaneous melanoma is one of the most lethal tumors among skin cancers and its incidence is rising worldwide. Recent data support the role of microRNAs (miRNAs) in melanoma carcinogenesis and their potential use as disease biomarkers. The aim of this study was to evaluate if miR-21-5p and miR-146a-5p expression is associated with melanoma histopathologic features including Breslow thickness, histological subtype, ulceration and regression status, and mitotic index. Our results showed that the combination of miRNAs and prognostic features can better differentiate cutaneous melanoma prognostic groups, considering overall survival and time-to-relapse. Our findings support the advantage of using molecular and diagnostic features in melanoma prognostication.

**Abstract:**

Background: Cutaneous melanoma (CM) is one of the most lethal tumors among skin cancers and its incidence is rising worldwide. Recent data support the role of microRNAs (miRNAs) in melanoma carcinogenesis and their potential use as disease biomarkers. Methods: We quantified the expression of miR-146a-5p and miR-21-5p in 170 formalin-fixed paraffin embedded (FFPE) samples of CM, namely 116 superficial spreading melanoma (SSM), 26 nodular melanoma (NM), and 28 lentigo maligna melanoma (LMM). We correlated miRNA expression with specific histopathologic features including Breslow thickness (BT), histological subtype, ulceration and regression status, and mitotic index. Results: miR-146a-5p and miR-21-5p were significantly higher in NM compared to SSM and LMM. The positive correlation between miR-146a-5p and miR-21-5p expression and BT was confirmed for both miRNAs in SSM. Considering the ulceration status, we assessed that individual miR-21-5p expression was significantly higher in ulcerated CMs. The increased combined expression of the two miRNAs was strongly associated with ulceration (*p* = 0.0093) and higher mitotic rate (≥1/mm^2^) (*p* = 0.0005). We demonstrated that the combination of two-miRNA expression and prognostic features (BT and ulceration) can better differentiate cutaneous melanoma prognostic groups, considering overall survival and time-to-relapse clinical outcomes. Specifically, miRNA expression can further stratify prognostic groups among patients with BT ≥ 0.8 mm but without ulceration. Our findings provide further insights into the characterization of CM with specific prognostic features. The graphical abstract was created with BioRender.com.

## 1. Introduction

The incidence of cutaneous melanoma (CM) has risen in the past decades, accounting for up to 5% of all skin tumors, and up to 65% of skin-cancer associated deaths [1,2]. The pathogenesis of CM is complex. Recognized risk factors include patient-related parameters such as fair phototypes and multiple dysplastic nevi, and genetic factors such as a positive family/personal history of melanoma [3]. A family history of CM poses the highest risk for the development of melanoma [4,5] and germline mutations in some high-penetrance CM susceptibility genes have been described, namely in *CDKN2A*, *CDK4*, *MITF*, *TERT*, *BAP1*, and *POT1* [6,7]. *CDKN2A* mutations are the most frequent in melanoma-prone families (20–40%) and affect the regulation of cell cycle by inhibition of CDK4 [8,9,10,11]. Also, *CDK4* can be found mutated in about 1.5% of melanoma-prone families [9,12,13,14]. *MITF* is a main regulator of differentiation, proliferation, and survival of melanocytes [9,14,15,16,17,18]. *BAP1* mutations impair DNA damage repair [9,12,14,19,20], while mutated *TERT* [9,14,21] and *POT1* [9,12,14,22] contribute to telomere elongation and maintenance, respectively. Environmental factors such as UV exposure play an important role in melanoma onset [6,23,24]. In fact, CM onset is closely linked to cumulative sun damage (CSD) produced by chronic or intermittent exposure to UV, and CMs have been classified in several categories according to this characteristic (high CSD and low CSD) in the WHO 2018 classification [25,26]. This classification includes also the melanoma subtypes described in the traditional clinicopathological classification scheme for melanoma, which are based on the pioneering work of Wallace Clark, Vincent McGovern, Martin Mihm, Richard Reed, and others in the late 1960s and 1970s [27,28,29,30]. Specifically, according to the clinical and histopathological features, invasive cutaneous melanoma has been classified into three major histological subtypes: superficial spreading melanoma (SSM), nodular melanoma (NM), and lentigo maligna melanoma (LMM) [30,31]. In the latest WHO classification, among melanoma arising on sun exposed skin, SSM are classified as low-CSD melanomas, while LMM fall into the group of high-CSD melanomas. The other remaining common subtype of CM, namely NM, can show characteristics to both the low- and the high-CSD types [32].

SSM accounts for 41% of CM cases and appears as a pigmented macule > 6 mm with irregular borders. SSM frequently localizes in sites prone to intense, sporadic sun-exposure, such as the trunk and extremities [31,33]. SSM is characterized by mutations in *BRAF*, *NRAS*, and less frequent in *MAP2K1*, *CTNNB1*, *PRKCA APC*, *BAP1*, and *PRKAR1A* genes in initial stages; in *TERT*, *CDKN2A*, *TP53*, and *PTEN* during malignant transformation, while during the metastatic phase can be found duplications of entire genome, copy number variations (CNVs), and aneuploidy [25,32,34,35]. NM accounts for 16% of CMs and is often symmetric, elevated, small in diameter, and has a single color. NM mostly localizes on the head and neck and is commonly observed in men older than 50 years. Compared to the other subtypes, NM possesses an increased growth rate, a more aggressive behavior, and an increased number of mitoses [31,36]. The genetic profile of NM exhibits similar frequencies of *BRAF* mutations to SSM; however, it has been revealed to present significantly higher frequencies of *NRAS* mutations [25,32,35,37]. LMM ranges from 5 to 15% of CM and consists of an irregular brown macule on chronically sun-damaged skin, such as the head and neck. LMM grows slowly over years [38]. LMM is characterized by inactivating mutations in NF1, copy number increases of *CCND1*, activating mutations of KIT, inactivating mutations of *TP53* and *ARID2*, and *TERT* promoter mutations [25,32,35,39,40].

Specific features have been previously reported as associated with a worse CM prognosis. The current staging system for CM (American Joint Committee on Cancer—AJCC, 8th edition, 2017) [1,31,41] proposed important markers of worse prognosis, including factors required for prognostic stage grouping, such as Breslow thickness (BT ≥ 0.8 mm) and ulceration, and factors recommended for clinical care, namely number of mitosis/mm^2^, Clark’s level, tumor infiltrating lymphocytes (TILs), lymphovascular invasion, and neurotropism [1,31,41]. Other important clinical and histopathologic parameters associated with worse prognosis are melanoma histological subtype, namely NM, and positive sentinel lymph node [42,43,44,45].

In the last twenty years, a great effort has been dedicated to the study of non-coding RNAs, especially microRNAs (miRNAs) in human cancers, including melanoma [46,47]. miRNAs dysregulation is involved in CM onset, growth, and metastatic spreading [46,48,49,50,51,52,53,54,55]. MiRNAs regulate multiple and specific target genes, determining an oncogenic or tumor-suppressive function. Oncogenic miRNAs (oncomiRs) target and downregulate tumor suppressor genes, while specific miRNAs play a protective role downregulating genes related to the neoplastic process: the altered balance of these two categories plays a pivotal role in the neoplastic process [56,57,58]. MRNAs are released inside extracellular vesicles [59,60,61]. MiRNA expression can be easily measured in archived formalin-fixed paraffin embedded (FFPE) tissues [62] and the miRNA global expression profile faithfully classifies normal and pathological cells and tissue [63,64].

Our group previously reported that miR-21-5p and miR-146a-5p expression significantly correlate with BT in SSM and their measurement could represent an objective molecular assay to support standard BT evaluation [65]. MiR-146a-5p and miR-21-5p promote transformation, proliferation, and invasion in melanoma [46,66,67]. In this study, we quantified the individual and combined expression of miR-146a-5p and miR-21-5p in an extended group of CM specimens analyzing their correlation with clinically relevant histologic features, including CM subtype, ulceration, regression, and mitotic index in order to evaluate their association and potential impact on patient overall survival (OS) and time-to-relapse (TTR).

## 2. Materials and Methods

### 2.1. Patients and Clinical Sample

Archive formalin-fixed paraffin embedded (FFPE) slices of 170 samples from patients with cutaneous melanoma were provided by Dermatopathology unit of Bologna University Hospital. Only samples with confirmed diagnosis of primary cutaneous melanoma were included in this study. Histopathological parameters (Breslow thickness, mitosis, ulceration, and regression) were evaluated using the American Joint Committee on Cancer—AJCC, 8th edition [1,2,68]. A total of 3 benign nevi (BN) were used as controls.

Acral lentiginous melanoma, melanoma in situ, and melanoma on nevus were excluded. 

Samples (*n* = 117) previously analyzed in Dika et al. are also included [65].

All subjects gave their voluntary and informed consent for use of clinical–pathological data and samples, and data privacy. The study was conducted in accordance with the Declaration of Helsinki, and the protocol was approved by the Ethics Committee CE-AVEC (417/2018/Sper/AOUBo).

Tissue sample examination was performed by expert dermatopathologists who selected only the area with morphological defined cancer cells from a 3 µm stained hematoxylin–eosin (HE) slide. 

### 2.2. RNA Extraction

For each patient 5 to 6 FFPE, 10 µm thick tissue sections were provided. According to the manufacturer’s instructions, RNA was extracted from selected CM cells area using miRNeasy FFPE kit (Qiagen, catalog number 217504). Deparaffinization was performed with xylene followed by an ethanol wash. An RNA quality check was measured by NanoGenius Spectrophotometer. Contamination of protein (260\280 ratio) was also checked. RNA was stored at −80 °C.

### 2.3. miRNA Expression Analysis

After total RNA extraction, 10ng RNA was converted to cDNA using miRCURY LNA RT kit (Qiagen, catalog number 339340) following the company’s guidelines. Real-time quantitative PCR (RT-qPCR) was performed using 4 μL of cDNA for each sample, 5 μL of miRCURY LNA SYBER Green PCR mix (Qiagen, catalog number 339346), and 1 μL of primer mix, namely primers for has-miR-146a-5p (Qiagen, catalog number YP00204688), fhashsa-miR-21-5p (Qiagen, catalog number YP00204230), and for SNORD44 (Qiagen, catalog number YP00203902). Each assay was tested in triplicate under the following thermal cycling conditions: 95 °C for 10 min, 40 cycles of 95 °C for 10 s, 60 °C for 1 min. Raw Cq values were obtained from BioRad CFX Maestro 2.2 software (version 5.2.008.0222). The calculation of relative expression was performed using 2^−ΔCt^ method [69] and SNORD44, a small nucleolar RNA, was used as a reference gene. Combined miRNA expression was calculated as miR-21-5p and miR-146a-5p normalized average expression.

### 2.4. Statistical Analysis

Clinical-histopathological features were summarized in a descriptive analysis considering the information provided by dermatologists for each patient. The association between miRNA expression and BT was investigated with Pearson’s correlation (r) and linear regression analysis. For all categorical variables, differences across the groups were analyzed using a Mann–Whitney test or unpaired *t*-test according to D′Agostino–Pearson Test and Shapiro–Wilk Test for normality. A two-sided *p*-value < 0.05 was considered significant. 

Overall survival was defined as the time elapsed between diagnosis and the death for any cause or the last available follow-up. Time-to-relapse was defined as the time elapsed between diagnosis and relapse. The survival curves were estimated by the Kaplan–Meier method, and the log-rank test was performed to test differences between the survival curves. Statistical analysis was performed using GraphPad Prism 8 (GraphPad Prism Software).

## 3. Results

We performed a molecular investigation to confirm the prognostic role of miR-146a-5p and miR-21-5p individual and combined expression in cutaneous melanoma. Combined miRNA expression was calculated as miR-21-5p and miR-146a-5p normalized average expression. For this purpose, 170 CM FFPE samples were collected, and their histological and clinical characteristics are summarized in Table 1. We tested the association of individual and combined miRNA expression of miR-146a-5p and miR-21-5p in all 170 CMs with melanoma histopathological features that are relevant in melanoma prognosis, including subtype, Breslow thickness, ulceration, mitotic rate, and regression.

### 3.1. miR-146a-5p and miR-21-5p Are More Expressed in NM Subtype 

The 170 CMs were classified based on their histological subtype as SSMs (N = 116), NMs (N = 26), and LMMs (N = 28). We quantified and compared the expression of miR-146a-5p and miR-21-5p in the three different subtypes. miR-146a-5p and miR-21-5p were significantly higher in NM compared to SSM and LMM. The expression was also tested in three benign nevi (BN) as a reference for miRNA expression. The expression of the miRNAs was lower in BN compared to melanomas. Also, evaluating the combined expression, NM presented a higher expression compared to SSM (*p* = 0.0012) and LMM (*p* = 0.0018). No statistically significant differences in miRNA expression were found between SSM and LLM subtypes (*p* = 0.3163) (Figure 1).

### 3.2. miR-146a-5p and miR-21-5p Expression Correlates with Breslow Thickness Only in SSM Histological Subtype

We analyzed and confirmed that miR-146a-5p and miR-21-5p expression correlates with BT in melanoma, independently from the subtype (Figure 2). When we tested the correlation between miRNA expression and BT in the three histological subtypes, we observed that miRNA expression correlated with BT only in the SSM histological subtype (Figure 2). In both NM and LMM, the correlation was not significant (Appendix A). 

Since a BT ≥ 0.8 mm is associated with a worse prognosis, we investigated the association of miRNA expression and BT ≥< 0.8 mm in all CMs and CM subtypes. The results confirmed that there was a significant difference in miRNA expression in all CMs (*p* < 0.0001) and in the SSM subtype (*p* < 0.0001) in the two prognostic groups (Figure 3 and Appendix A).

### 3.3. miR-146a-5p and miR-21-5p Expression Is Associated with Ulceration in Melanoma

Considering the ulceration status, we observed that miR-21-5p expression was significantly higher in ulcerated (*n* = 23) compared to non-ulcerated (*n* = 143) CMs (*p* = 0.0031). The combined expression was also higher in ulcerated samples (*p* = 0.0093). Instead, the increased expression of miR-146a-5p in ulcerated melanomas was borderline significant (*p* = 0.0513) (Figure 4). No significant difference was observed across the CM subtypes (Appendix A).

### 3.4. miR-146a-5p and miR-21-5p Expression Is Associated to Melanoma Mitotic Index in SSM Subtype

We tested the individual and combined miRNA expression in CM with available mitotic rate value (*n* = 161). Specifically, 108 CMs had a mitotic index <1/mm^2^ and 56 CMs have ≥1 mitosis/mm^2^. A significantly higher miRNA expression was observed in CM with higher mitotic rate (≥1/mm^2^) (*p* = 0.005 for combined and miR-21-5p expression, *p* = 0.0325 for miR-146a-5p expression) (Figure 5). This analysis was also performed in the different subtypes (81 SSM with <1 mitosis/mm^2^ vs. 34 SSM ≥ 1 mitosis/mm^2^, 7 NM with <1 mitosis/mm^2^ vs. 17 NM ≥ 1 mitosis/mm^2^ and 20 LMM with <1 mitosis/mm^2^ vs. 5 LMM ≥ 1 mitosis/mm^2^). Data showed that miRNA expression was significantly higher in SSM with a higher mitotic rate (Appendix A).

### 3.5. LMM Present Statistically Significant Different miRNA Expression Based on Regression Status

We did not observe any difference in miRNAs expression when we considered all subtypes together and their regression status (100 CM without regression vs. 65 CM with regression). However, when stratifying CM based on histological subtypes, individual and combined expression of miR-21-5p and miR-146a-5p were significantly higher in LMMs with regression (*n* = 18) compared to LMMs without regression (*n* = 7) (Figure 6). 

### 3.6. miR-146a-5p and miR-21-5p Expression and Prognostic Features Can Be Integrated to Improve Outcome Prediction

The combined miR-21-5p and miR-146a-5p expression and melanoma prognostic features, including subtype, BT, mitotic rate, ulceration, and regression were used in univariate analysis, alone or in combination, to test their association with overall survival (OS) and for time-to-relapse (TTR).

Our data confirm that the NM subtype was the most aggressive melanoma compared to SSM and LLM. Specifically, OS and TTR were significantly shorter in this subtype (*p* < 0.0001 for both OS and TTR) (Figure 7A). Also as expected, CMs with depth ≥ 0.8 mm had a worse prognosis (*p* < 0.0001) (Figure 7B). As we previously demonstrated in a smaller cohort of CMs, the combined miR-21-5p and miR-146a-5p expression that corresponds to 1.5 (the value corresponding to BT = 0.8 mm) [65] was used to dichotomize patients and perform survival analysis in all subtypes combined. CMs with expression of the two miRNAs higher than ≥1.5, displayed lower OS (*p* = 0.0091) and TTR (*p* = 0.0008) (Figure 7C). Also, ulceration (Figure 7C) and mitotic index (Appendix A) were confirmed to be prognostic factors. On the other hand, regression status did not show any statistically significant difference (Appendix A).

The combined miRNA expression was integrated with BT and ulceration in survival analysis. The results showed that for patients with BT ≥ 0.8 mm and without ulceration, a value of combined miRNA expression lower than 1.5 identified a favorable subgroup of CM patients, both in terms of time-to-relapse and overall survival (Figure 8).

## 4. Discussion

In the present study, we investigated the expression of miR-146a-5p and miR-21-5p in CM and their association with BT and other histopathologic prognostic parameters. In this study, we investigated the miRNA expression in all the three main CM subtypes: SSM, NM, and LMM.

High levels of miR-146a-5p and miR-21-5p are indicators of aggressivity in melanoma since they promote transformation, proliferation, and invasion. MiR-146a-5p has been defined as one of the most upregulated miRNAs in CM tumoral samples with respect to contiguous healthy tissue [70]. MiR-146a-5p acts targeting the NUMB gene, a repressor of NOTCH signaling, thus contributing to melanocytes transformation to primary melanoma [71]. MiR-146a-5p controls the activity of the MAPK signaling pathway, which has a key role in CM tumorigenesis, and it resulted to be differentially expressed in metastatic vs. primary CM [70]. Furthermore, miR-146a-5p increases cell migration and invasion in melanoma by directly targeting SMAD4 [46]. 

MiR-21-5p is highly expressed in melanoma and plays a central role in melanoma growth and progression [72,73]. Levels of miR-21-5p are significantly higher in primary melanoma tissues as compared to benign nevi [74] and in melanoma cell lines with respect to the normal melanocytes [72]. Acting on Sprouty, miR-21-5p sustains the cascade signaling of the BRAF/N-RAS pathway [75]. Moreover, miR-21-5p targets the anti-proliferative factor BCL2 and other tumor suppressor genes like PDCD4 and PTEN [76,77]. Therefore, the upregulation of miR-21-5p enhances the PI3K/AKT pathway. Knockdown of this miRNA reduces cell proliferation and promotes apoptosis [77].

We previously demonstrated a direct correlation between the combined expression of these two miRNAs and BT in the SSM subtype [65]. Here, we investigated also other histological subtypes, namely NM and LMM, and included all the main prognostic characteristics of melanoma to evaluate the prognostic performance of their combined assessment.

First, we demonstrated that the expression of these two miRNAs was significantly higher in NM compared to both SSM and LMM, thus confirming the more aggressive nature of this subtype. The worst prognosis of NM patients was also confirmed in our cohort by analyzing the overall survival of patients stratified by histological subtype [37].

Breslow thickness is the most important prognostic factor in CM [78]. In this study, we confirmed the correlation between miRNA expression and BT in SSM, while both NM and LMM did not presented any association between miRNA expression and BT. Considering the NM subtype, a possible explanation is that its aggressiveness could be relatively independent from BT, as recently described [79,80], and be associated with specific genetic alterations still to be identified. Considering all CM subtypes, we showed that a high expression of the combined miRNAs (≥1.5) correlates with a worse prognosis both in terms of OS (*p* = 0.0039) and TTR (*p* = 0.0001).

Ulceration is defined as full-thickness loss of undamaged epidermis overlying the primary melanoma tissue, with an associated host reaction [81]. This parameter is considered an adverse prognostic factor, and it has been classified as an independent prognostic factor defining the T-stage in the American Joint Committee of Cancer (AJCC) melanoma staging criteria VIII edition [41,82]. In our cohort of CMs (all subtypes combined), we confirmed the prognostic role of ulceration: survival was significantly shorter in ulcerated CMs (OS and TTR; *p* value < 0.0001). Some miRNAs have been described to be associated with ulceration. DiVincenzo et al. performed a high-throughput analysis using Nanostring technology and they found 24 miRNAs dysregulated in melanoma tissue with ulceration compared with non-ulcerated tumors [83]. In particular, they showed that ulcerated melanoma tissue had a 2.5-fold increase in miR-146a-5p expression compared to non-ulcerated melanoma. Our study strengthens their findings; indeed, the combined and individual miRNA expression resulted differently expressed in ulcerated vs. non-ulcerated CMs.

Although not included in the latest AJCC staging system, we also tested the association of our prognostic miRNAs with mitotic rate and regression. Our data showed that a mitotic rate greater than 1/mm^2^ was associated with worse prognosis (OS and TTR). A recent study proposed that a mitotic rate of a mitotic rate ≥ 2 mitoses/mm^2^ should be considered as a more accurate prognostic factor than ≥1 mitosis/mm^2^, particularly in CM with a BT ≥ 0.8 mm [84]. Here, we showed that mitotic index is correlated with a differential miRNA expression in CM, specifically in the SSM subtype. In melanocytic lesions with ≥1/mm^2^ mitosis, the level of miR-21-5p is higher than in lesions without mitotic activity. This observation confirms the pivotal role of this miRNA in promoting cancer proliferation [74]. 

Regression of CM was reported at histological evaluation in up to 1/3 of the specimens and is clinically characterized by hyperpigmentation, followed by depigmentation of part of or the entire lesion, resulting in blue, pink, white, or gray areas. Regression can be defined as partial, segmental, or complete replacement of melanoma cells with a variable host response. Specifically, it consists of a host reaction with lymphocyte and macrophage infiltration, with the absence of nests of melanocytes in the atrophic epidermis, but the presence of thin dermal collagen [85]. Regression is under investigation as a prognostic factor in melanoma. It has been considered an adverse prognostic element for a long time: Sook Jung Yun et al. demonstrated that regression in melanomas with a radial growth adjacent to a vertical growth of the tumor is associated with a lymphatic invasion [86]; but recent studies highlighted a possible favorable role, suggesting an immune-mediated response of activated T-lymphocytes against neoplastic cells [41]. A recent systematic review on this topic revealed that the chance to detect a positive sentinel lymph node was significantly lower in those subjects with primary CM with regression compared to those without [87]. Moreover, it was demonstrated that regression was found more frequently in thin CMs and in the LMM subtype [88]. From our analysis, CMs with regression seem to have a better survival, however, there was no statistically significant difference. In our series, the expression of miR-146a-5p and miR-21-5p did not show a significant difference in CMs with/without regression, except for LMM; an increased expression of combined and individual miR-21-5p and miR-146a-5p was observed in LMM with regression, suggesting that a correlation with the inflammatory state and microenvironment in LMM should be further investigated. The low number of analyzed cases, though, does not permit us to draw any conclusions.

Based on these observations, we tested if the addition of miR-146a-5p and miR-21-5p molecular measurement to BT and ulceration status could be of use in patient prognostication. The integration of BT, ulceration, and miRNA expression provided interesting results. In the subgroup of CMs with BT > 0.8 mm and without ulceration, tumors presenting a combined miRNA expression <1.5 had a shorter overall survival and TTR compared those with miRNA ≥ 1.5, suggesting a potential contribution as independent prognostic factor. 

## 5. Conclusions

In conclusion, our study confirms the clinical usefulness of testing miR-146a-5p and miR-21-5p expression in CM and its role in identifying patients with a favorable prognosis among those with adverse prognostic factors, including tumor depth and ulceration that are relevant in CM prognosis. These findings provide further insights for the diagnosis and treatment of CM with specific adverse prognostic features and represent a promising prognostic biomarker even for early-stage disease. A multicentric study should be conducted to validate these results in a larger cohort of patients.

## 6. Patents

The work reported in this manuscript is covered by an Italian patent (n. 102018000009148 granted on 08/09/2020).

## Figures and Tables

**Figure 1 cancers-16-01688-f001:**
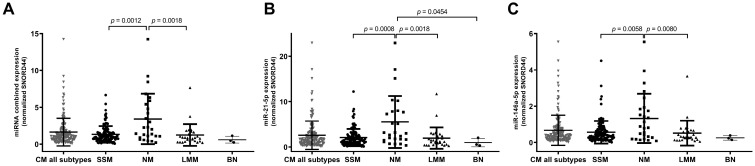
Association of the combined and individual miR-21-5p and miR-146a-5p expression with melanoma histotypes. Figures show the scatter plot distribution of combined miRNA expression (**A**) and individual miR-21-5p (**B**) and miR-146a-5p (**C**) in all cutaneous melanomas (CM) and in the three main histological subtypes, namely superficial spreading melanoma (SSM), nodular melanoma (NM), and lentigo maligna melanoma (LMM). miRNA expression was also tested in three benign nevi (BN) as a reference. Combined and individual miRNA expression is higher in NM compared to SSM and LMM.

**Figure 2 cancers-16-01688-f002:**
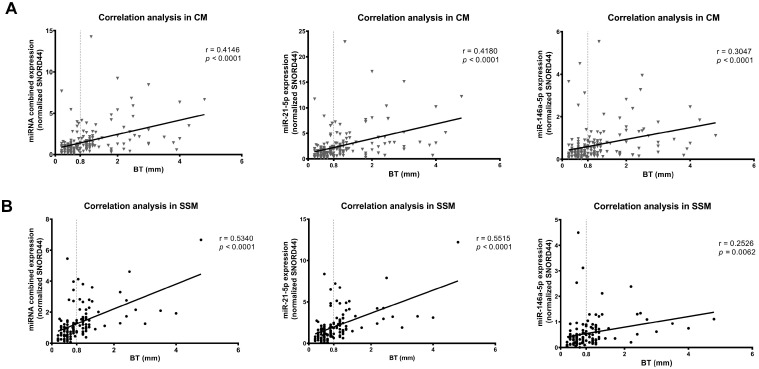
Correlation of combined and individual miRNA expression with Breslow thickness (BT) in all melanomas (CM) and superficial spreading melanoma (SSM) subtype. Figures show the correlation plot of the combined and individual miRNA values and BT values from the same tumor. A significant positive correlation in all CMs (**A**) and SSM subtype (**B**) is reported (Pearson r). Simple linear regression line is shown in the graphs.

**Figure 3 cancers-16-01688-f003:**
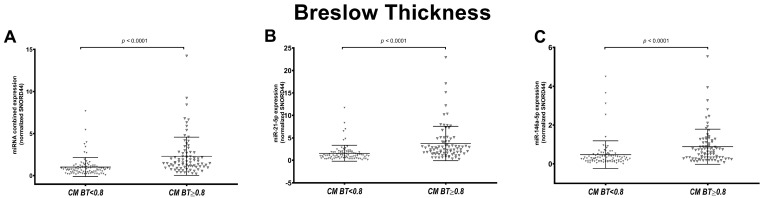
Association of combined and individual miR-21-5p and miR-146a-5p expression with Breslow thickness (BT). Figures show the scatter plot distribution of the combined miRNA expression (**A**) and individual miR-21-5p (**B**) and miR-146a-5p (**C**) in all cutaneous melanomas (CMs) grouped based on BT: CMs with BT < 0.8 mm and CMs with BT ≥ 0.8 mm. Combined and individual miRNA expression was significantly higher in CMs with BT ≥ 0.8 mm compared to CMs with BT < 0.8 mm.

**Figure 4 cancers-16-01688-f004:**
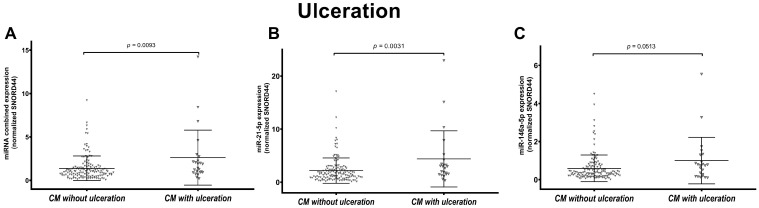
Association of combined and individual miR-21-5p and miR-146a-5p expression with ulceration status. Figures show the scatter plot distribution of the combined miRNA expression (**A**) and individual miR-21-5p (**B**) and miR-146a-5p (**C**) in all cutaneous melanomas (CMs) grouped based on ulceration status: CMs without ulceration and CMs with ulceration. Combined and individual miR-21-5p expression was significantly higher in ulcerated CMs.

**Figure 5 cancers-16-01688-f005:**
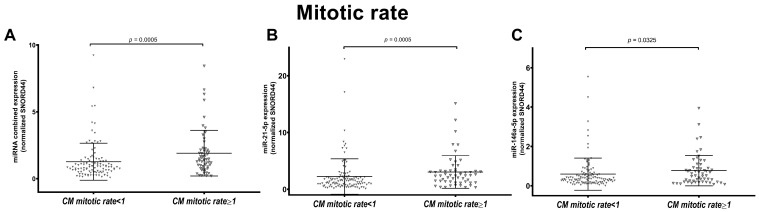
Association of combined and individual miR-21-5p and miR-146a-5p expression with mitotic rate. Figures show the scatter plot distribution of the combined miRNA expression (**A**) and individual miR-21-5p (**B**) and miR-146a-5p (**C**) in all cutaneous melanomas (CMs) grouped based on mitotic rate: CMs with <1 mitosis/mm^2^ and CMs with ≥1 mitosis/mm^2^. Combined and individual miRNA expression was higher in CMs with ≥1 mitosis/mm^2^ compared to CMs with <1 mitosis/mm^2^.

**Figure 6 cancers-16-01688-f006:**
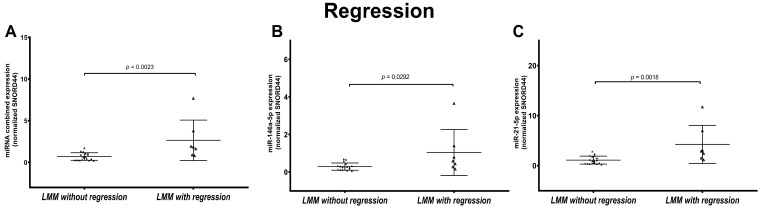
Association of combined and individual miR-21-5p and miR-146a-5p expression in lentigo maligna melanoma (LMM) based on regression status. Figures show the scatter plot distribution of the combined miRNA expression (**A**) and individual miR-21-5p (**B**) and miR-146a-5p (**C**) LMMs grouped based on regression status: LMMs without regression and LMMs with regression. Combined and individual miRNA expression was significantly higher in LMMs with regression.

**Figure 7 cancers-16-01688-f007:**
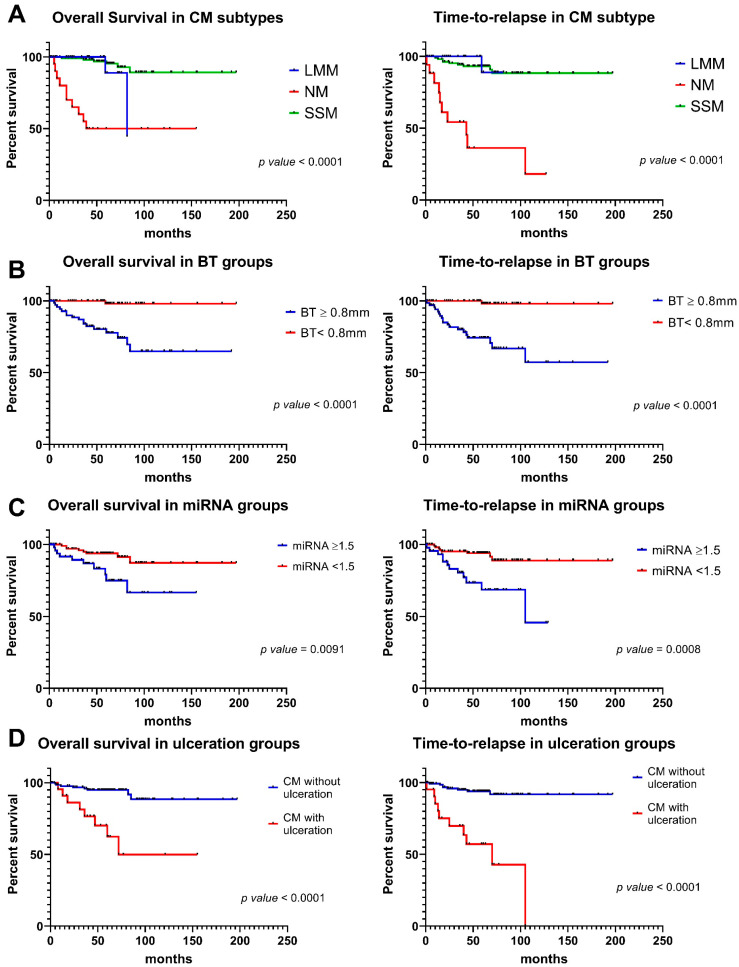
Kaplan-–Meier plots for overall survival (OS) and time-to-relapse (TTR) in CM prognostic groups. (**A**) The plots show worse OS and TTR curves in patients with nodular melanoma (NM) histotype compared to superficial spreading melanoma (SSM) and lentigo maligna melanoma (LMM). (**B**) The plots show significantly worse OS and TTR curves in patients with BT ≥ 0.8 mm versus those with BT < 0.8 mm. (**C**) The plots show significantly worse OS and TTR curves in patients with miRNA expression ≥1.5 versus those with miRNA < 1.5. (**D**) The plots show significantly worse OS and TTR curves in patients with ulceration compared to patients without ulceration.

**Figure 8 cancers-16-01688-f008:**
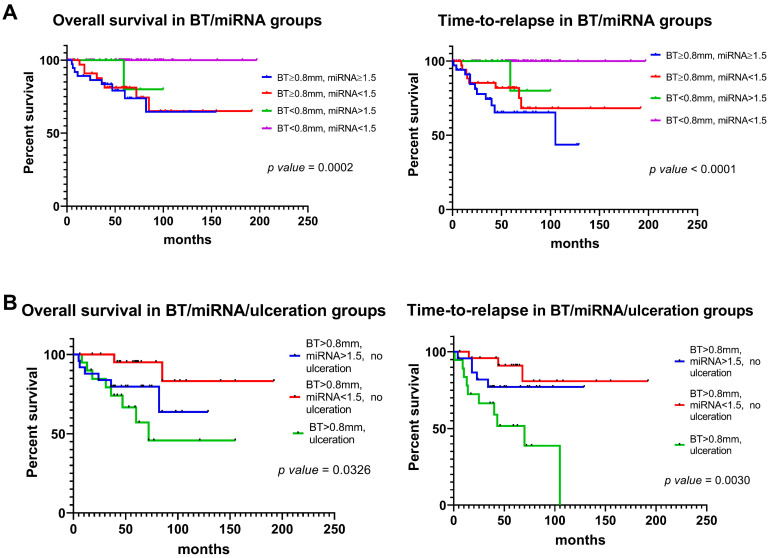
Kaplan–Meier plots for overall survival (OS) and time-to-relapse (TTR) in patients grouped according to miRNA expression, Breslow thickness (BT), and ulceration status. (**A**) The plots show significantly worse OS and TTR curves in patients with BT ≥ 0.8 mm versus those with BT < 0.8 mm. Among melanomas with BT ≥ 0.8 mm, those with miRNA expression ≥1.5 have a worse TTR compared to those with miRNA expression < 1.5. (**B**) Among patients with BT ≥ 0.8 mm, ulcerated melanomas have the worst prognosis in terms of OS and TTR. Among patients with BT ≥ 0.8 mm and without ulceration, a worse prognosis is showed by patients with miRNA expression ≥1.5 compared to those with miRNA expression < 1.5.

**Table 1 cancers-16-01688-t001:** Clinical and tumor features of patients with cutaneous SSM, LMM, and NM.

	SSM	LMM	NM	All Samples
Total (n°)	116	28	26	170
of which from Dika et al. [39] ^1^	90	2	25	117
Gender (n°)				
Male	64	13	20	97
Female	52	15	6	73
Age				
<50 years	24	2	8	34
≥50 years	92	26	18	136
Breslow Thickness (n°)				
<0.8 mm	65	23	1	89
≥0.8 mm	51	5	25	81
Ulceration (n°)				
Presence	14	0	9	23
Absence	102	26	15	143
Not available		2	2	4
Mitosis (n°)				
<1/mm^2^	81	20	7	108
≥1/mm^2^	34	5	17	56
Not available	1	3	2	6
Regression (n°)				
Presence	54	7	4	65
Absence	61	18	21	100
Not available	1	3	1	5

^1^ Samples from patients already analyzed in Dika et al. [39]. Abbreviations: SSM, superficially spreading melanoma; LMM, lentigo maligna melanoma; NM, nodular melanoma.

## Data Availability

The data that support the findings of this study are available upon reasonable request.

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
