# Peer review of "Association of miR-146a-5p and miR-21-5p with Prognostic Features in Melanomas"

_cancers, 2024, doi:10.3390/cancers16091688_

Round 1

Reviewer 1 Report

Comments and Suggestions for Authors

The authors correlated different clinical features and tumor grades to the presence of miR-21-5p and miR-146a-5p.  This paper was descriptive in nature and does not present cause and effect.

Major correction:  The authors are missing controls in these experiments as they did not look at the microRNA levels studied in the normal tissue adjacent to the tumor.  These findings would be more meaningful if the levels of microRNAs were also evaluated in wild-type skin.

The authors should try to knock down these microRNAs in melanoma cell lines and determine if the cells become less mitotic and revert to wild-type (or less severe phenotypes).

There is no Table 1.

Minor corrections:  Please remove the titles for the correlation graphs/scatter plots for figures 2-8 as they are not needed due to the figure title and the axes labels.

Comments on the Quality of English Language

There are a few grammatical errors in this manuscript.  Please proofread.  For example, line 53 should read has risen not has raised.

Author Response

The authors correlated different clinical features and tumor grades to the presence of miR-21-5p and miR-146a-5p. This paper was descriptive in nature and does not present cause and effect. Major correction:

-The authors are missing controls in these experiments as they did not look at the microRNA levels studied in the normal tissue adjacent to the tumor. These findings would be more meaningful if the levels of microRNAs were also evaluated in wild-type skin.

Reply: We don’t understand what the reviewer refers to as wild-type skin, but the studies comparing the miRNome profile of melanoma and normal tissue usually use benign nevi (Torres R, Lang UE, Hejna M, Shelton SJ, Joseph NM, Shain AH, Yeh I, Wei ML, Oldham MC, Bastian BC, Judson-Torres RL. MicroRNA Ratios Distinguish Melanomas from Nevi. J Invest Dermatol. 2020 Jan;140(1):164-173.e7. doi: 10.1016/j.jid.2019.06.126. Epub 2019 Sep 30. PMID: 31580842; PMCID: PMC6926155). To answer the reviewer request, we included benign melanocytic tumors (benign nevi). As a reference for miRNA expression miRNA expression. miR-21-5p and miR-146a-5p in benign nevi was added in Figure 1.

The authors should try to knock down these microRNAs in melanoma cell lines and determine if the cells become less mitotic and revert to wild-type (or less severe phenotypes).

Reply: We thank to reviewer for his/her comment. The aim of this paper is to study the association of miRNA expression with melanoma prognostic biomarkers used in the clinical management of melanoma patients. We did not aim to provide the functional role of these miRNA, which are already described in literature (Dika E, Riefolo M, Porcellini E, Broseghini E, Ribero S, Senetta R, Osella-Abate S, Scarfì F, Lambertini M, Veronesi G, Patrizi A, Fanti PA, Ferracin M. Defining the Prognostic Role of MicroRNAs in Cutaneous Melanoma. J Invest Dermatol. 2020 Nov;140(11):2260-2267. doi: 10.1016/j.jid.2020.03.949. Epub 2020 Apr 8. PMID: 32275975.)

There is no Table 1.

Reply: We apologize for the missing table and we thank to reviewer for his/her comment. We added the missing Table.

Minor corrections: Please remove the titles for the correlation graphs/scatter plots for figures 2-8 as they are not needed due to the figure title and the axes labels.

Reply: We thank to reviewer for his/her comment. We added titles in the figures because being so similar to each other, the title allows the reader to better find and focus on the figure of interest.

Reviewer 2 Report

Comments and Suggestions for Authors

This study evaluated two miRNA in cutaneous melanoma. The quantity of these two miRNAs was correlated with several clinicopathological features. The manuscript is well written, it is easy to understand, and the data worth publishing. However, I am not sure why the typical tables of correlations between the miRNA levels with the clinicopathological variables, OS, PFS, univariate and multivariate are not shown.

 Additional comments:

 (1) In the abstract. Please describe the number of cases for each subtype (NM, SSM, LMN)

 (2) Regarding this sentence in the abstract “The positive correlation between miR-146a-5p and miR-21-5p expression and BT was confirmed for both miRNAs in an extended group of SSM”. What do you mean by “extended”?

 (3) Why are CDKN2A, CDK4, MITF, TERT, BAP1, and POT1 genes relevant in the pathogenesis of cutaneous melanoma?

 (4) Line 63. Are the 4 histological subtypes included in the current WHO classification?

 (5) Lines 66-76. Apart from the clinical differences, is there any biological difference between the 4 histological subtypes? Any genomic or transcriptomic differences?

 (6) You may describe the Breslow and Clark classification in the appendix or supplementary data.

 (7) Is the 0.8mm cutoff only for T1 cases?

 (8) In the introduction, what is the function of miR-146a-5p and miR-21-5p?

 (9) Why were acral lentiginous melanoma, melanoma in situ, and melanoma on nevus were excluded?

 (10) Please provide the catalog number of the miRNeasy.

 (11) Line 147. Why not using the Shapiro-Wilk test?

 (12) Line 159. Sorry, I cannot find Table 1 in the manuscript.

 (13) Apart from melanoma, did you quantify these miRNAs in normal skin?

 (14) How did you calculate the “combined expression”?

 (15) How was the mitotic rate evaluated?

(16) How do you define the “regression status”? (line 235)

 (17) Is it possible to make a table with the correlations of the OS and PFS? Univariate and multivariate analysis?

Author Response

This study evaluated two miRNA in cutaneous melanoma. The quantity of these two miRNAs was correlated with several clinicopathological features. The manuscript is well written, it is easy to understand, and the data worth publishing. However, I am not sure why the typical tables of correlations between the miRNA levels with the clinicopathological variables, OS, PFS, univariate and multivariate are not shown.

 Additional comments:

(1)    In the abstract. Please describe the number of cases for each subtype (NM, SSM, LMN).

Reply: We thank to reviewer for his/her suggestion. We added the number of cases for each subtype (lines 38-39)

(2)    Regarding this sentence in the abstract “The positive correlation between miR-146a-5p and miR-21-5p expression and BT was confirmed for both miRNAs in an extended group of SSM”. What do you mean by “extended”?

Reply: We apologize for the oversight. Since the correlation was performed on all SSM samples, we removed “extended”.

(3)    Why are CDKN2A, CDK4, MITF, TERT, BAP1, and POT1 genes relevant in the pathogenesis of cutaneous melanoma?

Reply: We added some information to explain why these genes are important for melanoma pathogenesis (lines 63-69):

(4)    Line 63. Are the 4 histological subtypes included in the current WHO classification?

Reply: We update the introduction part to better explained the inclusion of the histological subtypes (SSM, NM, LMM) in the current WHO classification (lines 70-82).

(5) Lines 66-76. Apart from the clinical differences, is there any biological difference between the 4 histological subtypes? Any genomic or transcriptomic differences?

Reply: We added genomic differences of SSM (lines 85-89), NM (lines 93-95) and LMM (lines 97-99).

(6) You may describe the Breslow and Clark classification in the appendix or supplementary data.

Reply: We added Breslow and Clark classification in the supplementary data.

(7)Is the 0.8mm cutoff only for T1 cases?

Reply: To better explain the role of 0.8mm cutoff, we added a Table with melanoma staging classification as supplementary information (supplementary data).

(8) In the introduction, what is the function of miR-146a-5p and miR-21-5p?

Reply: We added this information in introduction (lines125-126)

(9) Why were acral lentiginous melanoma, melanoma in situ, and melanoma on nevus were excluded?

Reply: On the basis of histopathological features of the intra-epidermal component of the tumour adjacent to any dermal invasive component the major histological subtype are SSM, LMM and NM. Since our main purpose was to consider the correlation with Breslow thickness, which is nowadays considered the most important prognostic factor for cutaneous melanoma, we focused on >pT1a melanomas.

(10) Please provide the catalog number of the miRNeasy

Reply: We added the missing information (line 153)

(11) Line 147. Why not using the Shapiro-Wilk test?

Reply: We used D'Agostino-Pearson Test, however, based on your suggestion, we used also Shapiro-Wilk test, which has been added in the material and methods part (line 176).

(12) Line 159. Sorry, I cannot find Table 1 in the manuscript.

Reply: We apologize for the missing table. We added the missing Table.

(13) Apart from melanoma, did you quantify these miRNAs in normal skin?

Reply: We don’t understand what the reviewer refers to as wild-type skin, but the studies comparing the miRNome profile of melanoma and normal tissue usually use benign nevi (Torres R, Lang UE, Hejna M, Shelton SJ, Joseph NM, Shain AH, Yeh I, Wei ML, Oldham MC, Bastian BC, Judson-Torres RL. MicroRNA Ratios Distinguish Melanomas from Nevi. J Invest Dermatol. 2020 Jan;140(1):164-173.e7. doi: 10.1016/j.jid.2019.06.126. Epub 2019 Sep 30. PMID: 31580842; PMCID: PMC6926155). To answer the reviewer request, we included benign melanocytic tumors (benign nevi). As a reference for miRNA expression miRNA expression. miR-21-5p and miR-146a-5p in benign nevi was added in Figure 1.

(14) How did you calculate the “combined expression”?

Reply: We thank to reviewer for his/her comment. We already reported this information in material and methods, however, we also added this part in the results (line 186-188)

(15) How was the mitotic rate evaluated?

Reply: Mitotic rate was evaluated using American Joint Committee on Cancer – AJCC, 8th edition, as described in the material and methods part (line 137-139). According to the European consensus-based interdisciplinary guideline for melanoma, the presence of >1/mm2 is considered as an adverse prognostic feature.

(16) How do you define the “regression status”? (line 235)

Reply: Regression is defined as the segmental replacement of melanoma by fibrosis, with increased vascularity and melanophages, and a lymphocytic infiltrate of variable density, with or without a residual epidermal component.

(17) Is it possible to make a table with the correlations of the OS and PFS? Univariate and multivariate analysis?

Reply: We did not use the multivariate analysis because we decided to combine the miRNA expression with the only two recognized prognostic parameters for melanoma (ulceration and thickness) generating the combination group. For this analysis there is no other parameter to include in a multivariate testing.

Round 2

Reviewer 1 Report

Comments and Suggestions for Authors

Thank you to the authors for addressing the main concern that there were no controls for the initial miRNA assessment.

Comments on the Quality of English Language

Minor edits to writing needed.